# The Impact of Positive and Adverse Experiences in Adolescence on Health and Wellbeing Outcomes in Early Adulthood

**DOI:** 10.3390/ijerph21091147

**Published:** 2024-08-29

**Authors:** Lynn Kemp, Emma Elcombe, Stacy Blythe, Rebekah Grace, Kathy Donohoe, Robert Sege

**Affiliations:** 1Centre for Transforming Early Education and Child Health, School of Nursing and Midwifery, Western Sydney University, Penrith Campus, Kingswood, NSW 2747, Australia; e.elcombe@westernsydney.edu.au (E.E.); stacy.blythe@uts.edu.au (S.B.); rebekah.grace@westernsydney.edu.au (R.G.); kathy.donohoe@westernsydney.edu.au (K.D.); 2Institute for Clinical Research and Health Policy Studies, Tufts Medical Center, Boston, MA 02111, USA; robert.sege@tuftsmedicine.org

**Keywords:** positive childhood experiences, adverse childhood experiences, adolescence, longitudinal studies

## Abstract

This study evaluated the associations between positive and adverse experiences and environments in adolescence and health, education and employment outcomes in early adulthood. Data were extracted from the Longitudinal Studies of Australian Youth cohort that commenced in 2003. The items were conceptually mapped to Positive and Adverse Youth Experiences and environments (PYEs and AYEs) at 15, 16 and 17 years old and outcomes at 25 years old. The associations between PYEs, AYEs and general health, mental health, education and employment were examined, including testing whether PYEs mitigated the association between AYEs and outcomes. A higher number of AYEs was associated with poorer health, education, and employment outcomes. Conversely, a higher number of PYEs was correlated with positive outcomes. The participants with higher PYEs had significantly greater odds of better general and mental health outcomes, even after accounting for AYEs. This relationship was not observed for employment or education outcomes. Adolescence and the transition to adulthood are critical developmental stages. Reducing adverse experiences and environments and increasing positive ones during adolescence could enhance adult wellbeing.

## 1. Introduction

There is extensive evidence that adverse childhood experiences in the early years (ACEs) have significant impacts on long-term health, development and economic outcomes in adulthood. Knowledge of the effects of ACEs began with the landmark 1998 study by Felitti et al. [1]. Subsequent research has confirmed these relationships and expanded our understanding of the number and types of adverse events and the mechanisms and pathways of impact [2,3,4].

More recently, attention has been turned to considering the impact of positive experiences on adult health and wellbeing, with evidence suggesting that people’s outcomes as adults are impacted by both adversity and the positive experiences they had in childhood: positive childhood experiences (PCEs). This evidence suggests that not only are PCEs important for positive outcomes in adulthood, particularly mental health, but that they also buffer, or mitigate, the negative impacts of ACEs [5,6,7]. Although there is no clear consensus on what constitutes PCEs, they have been conceptualised in the Healthy Outcomes from Positive Experiences (HOPE) framework as comprising four building blocks for key positive childhood experiences and the sources of those experiences and opportunities that help children grow into healthy, resilient adults. These building blocks consist of relationships within the family and with others; safe, equitable and stable environments at home and in school; social and civic engagement for belonging and connectedness; and opportunities for emotional growth for self-awareness and self-regulation [8,9].

Evidence regarding the mitigating potential of PCEs has largely been generated from cross-sectional studies of adults, mostly aged over 25 years, reporting their current health, wellbeing, relationships and economic outcomes, and recalling their childhood ACEs and/or PCEs [1,5,9,10]. Such cross-sectional, recall-based studies, however, are not able to draw causal inferences and may be subject to recall bias. Studies relying on a lesser period of recall have been undertaken more recently, exploring the relationships between ACEs and PCEs and outcomes in adolescence and early adulthood, including youth crime and recidivism [11,12]; child and youth depression and anxiety [13,14]; college student mental and emotional health [15]; psychological resilience [16]; prosocial behaviour [17]; chronic disease onset [18]; the use of maladaptive schemas in adolescence including emotional deprivation, subjugation, mistrust, feelings of abandonment and failure and social isolation [19]; and substance use [20].

This growing body of evidence presents a risk that ACEs and PCEs could be interpreted as deterministic of trajectories into adulthood, with considerable debate surrounding the utility and efficacy of ACE screening, in particular, at key adult transition times such as pregnancy or with parents of young children [21,22]. However, adolescence represents another important time of transition. Neurodevelopmental evidence suggests that the brain has a second peak of malleability at this time, particularly, prefrontal cortex pruning, which is associated with social, emotional and executive functioning development [23,24]. Adolescence is also a vulnerable period for psychological distress, associated with conflict with peers and adverse family circumstances [25]. This is a period where the impacts of daily stressors relating to peers and increasingly complex social situations and family relationships characterised by increasing independence and autonomy and conflict with parents become frequent and intense, in addition to academic demands [26,27]. It is also a period in which severe psychiatric conditions, including schizophrenia and eating disorders, often begin.

Despite adolescence being a developmental and life transition point, there has been very little attention paid to the impact on the adult outcomes of experiences during this life period, with research often focused on specific adverse experiences [28]. For example, a longitudinal study undertaken in Australia by Parker et al. [29] found that low school belonging at age 15 was associated with not being in education, employment or training at age 16–20, independent of school graduation; a longitudinal study in Taiwan found that homelessness and cumulative adverse experiences during adolescence negatively impacted education and work trajectories from age 18 to 22 [30]; and an analysis of the US National Longitudinal Study of Adolescent Health showed that material deprivation in adolescence was associated with poorer physical and mental health in adulthood [31]. An analysis of a longitudinal study in Sweden found that social anxiety in adolescence was associated with depressive symptoms in adulthood through a pathway of peer stress and stress associated with school performance and homelife [32], and the quality of peer relationships was associated with life satisfaction [33].

Notably, there is scant evidence on the impact of positive experiences in adolescence on outcomes and no research on the potential for positive experiences to mitigate adverse experiences at that life stage. The evidence above suggests that adverse and positive experiences that occur in adolescence (such as social anxiety or school belonging) may look different to, and may be additional to, those experienced in early childhood and the commonly measured ACEs (abuse, neglect and household dysfunction), and there have been no specific measures of adverse or positive adolescent experiences developed to date. Guo and colleagues [34] have used the Longitudinal Study of Australian Children to identify positive experiences for children aged 0–11 years and validate their efficacy in predicting impacts on mental health and school outcomes at age 14. However, the nature and impact of adverse and positive experiences in adolescence on early adult outcomes remains unknown. Such knowledge could assist services and those engaging with young people to implement policies and strategies to minimise adverse and promote Positive Youth Experiences to maximise health and productivity in adulthood.

### Hypotheses

In order to better understand the role of positive and adverse adolescent experiences on young adult health, educational and economic outcomes, we analysed data from the Longitudinal Studies of Australian Youth (LSAY). In this analysis, we categorised a range of adolescents’ adverse and positive experiences and examined their relationships to general health, mental health, and educational and economic outcomes in early adulthood (age 25). We hypothesised the following:Adverse Youth Experiences (AYEs) are associated with poorer general health, mental health, and educational and employment outcomes in young adulthood;Positive Youth Experiences (PYEs) are associated with better outcomes in young adulthood;Positive Youth Experiences (PYEs) mitigate or buffer the negative impact of Adverse Youth Experiences (AYEs) on young people’s outcomes in early adulthood in the same ways as those demonstrated in the cross-sectional studies of adult recall of their ACEs and PCEs.

## 2. Materials and Methods

### 2.1. Participants

This study used publicly available data from the Longitudinal Surveys of Australian Youth (LSAY) managed by the National Centre for Vocational Education Research and conducted by Wallis Social Research on behalf of the Australian Government Department of Education. The LSAY recruited and initially surveyed adolescents at around 15 years of age (school year 10) and then followed up every year until age 25. Between 1995 and 2015, the LSAY recruited a new cohort approximately every three years. The sample was drawn from Australian schools that participate in the Organisation for Economic Co-operation and Development (OECD) Programme for International Student Assessment. The LSAY aimed to recruit a nationally representative sample of over 10,000 adolescents in each wave who are in non-home-schooling education, who have had more than one-year of education in English and are able to undertake the assessment in English, and do not have cognitive, mental or physical disabilities that would preclude them from understanding and/or physically undertaking the assessment [35,36]. The LSAY focus is on youth school and life experiences and, later, educational and employment outcomes. The LSAY does not ask adolescents about their current or earlier life experiences of abuse, neglect or family dysfunction that are typically the subject of Adverse Childhood Experience measurement [1].

In order to identify the LSAY cohort suitable for this study, the LSAY questions asked of adolescents (15 to 17 years of age, waves 1 to 3) in all completed cohorts (recruited 1995 to 2012) were searched to find those questions which best aligned with stressors and with the four building blocks of the HOPE framework; (1) relationships within the family and with others; (2) safe, equitable and stable environments at home and in school; (3) social and civic engagement for belonging and connectedness; and (4) opportunities for emotional growth for self-awareness and self-regulation. The LSAY03 cohort was found to be the most recent cohort which asked the participants a selection of questions suitable for conceptually mapping to the HOPE framework. For example, a question asked at 15 years was ‘how important in your life are: Your close friends?’, which was mapped to the HOPE item ‘relationships with family and others’. The current study used the sub-sample of adolescents who commenced the LSAY in 2003 (LSAY03) and completed all four survey waves 1, 2, 3 and 10, administered at age 15, 16, 17 and 25, respectively.

### 2.2. Procedure

The conceptual mapping was based on literature describing key positive and adverse experiences in adolescence (see, for example, Núñez-Regueiro and Núñez-Regueiro [26]), including activities such as carer responsibilities that can adversely impact the young person’s attainment of increasing independence [37], and the study team’s understanding from working with vulnerable adolescents in a number of concurrent studies (see, for example, the ReSPECT project [38]) and their descriptions of adversities and positive experiences in their lives. Conceptual mapping was initially undertaken by the second author, with consensus reached after discussion amongst the study team. A complete list of included questions and their mapped domains is shown in the Appendix A.

Ten questions asked between the ages of 15 and 17 were used to measure each adolescent’s Positive Youth Experiences (PYEs). The questions included related to the adolescent’s relationships (building block 1), for example, ‘how important in your life are the family members you live with?’ (scale from 0 to 10); environment (building block 2), for example, ‘my school is a place where: I feel safe and secure’ (responses from ‘strongly agree’ to ‘strongly disagree’); and social engagement (building block 3), for example, ‘how often did you take part in the following school-organised activities: e.g., sport, music, debating, drama, peer support, school-sponsored volunteer activities?’ (responses from ‘at least once a week’ to ‘never’). No questions were identified in the surveys that aligned with the HOPE domain of emotional growth (building block 4).

Seven questions were used to assess the adolescents’ adverse experiences (Adverse Youth Experiences (AYEs)). These included poverty, for example, ‘are you happy with your standard of living?’ (responses ‘very happy’ to ‘very unhappy’); not living with parents or relatives; feeling alienated, for example, ‘school is a place where: I feel like an outsider’ (responses ’strongly agree’ to ‘strongly disagree’) and having carer responsibilities.

Outcome data were taken from LSAY03 wave 10 (25 years of age). The outcomes considered cover four key areas of adulthood: (1) educational attainment; (2) participating in employment, education or training; (3) general health; and (4) mental health. Six items asked at age 25 were used to assess the four primary outcome measures. Three questions were asked about work and study, including if the person was in part- or full-time employment, if they were enrolled in further education and if they were receiving government welfare. These questions were used to determine if the 25-year-old was in education, training or satisfactory employment. Responders who were either in full- or part-time study or were employed and not receiving welfare were considered to be in education, training or satisfactory employment. Responders who had completed a certificate IV (typically vocational or trade certificate) or higher (including university qualifications) were considered to have post-school education. Responders who reported their general health to be ‘Fair’ or ‘Poor’ were considered to have poor health while those answering ‘Good’, ‘Very good’ and ‘Excellent’ were considered to have good general health. Mental health was assessed using the Kessler Psychological Distress measure. LSAY asked six items from this tool (K6), including questions related to nervousness, hopeless, restlessness, effortfulness, sadness and worthlessness. As per the Australian scoring rubric, items were scored from 1 to 5, for a total score range of 6 to 30. Responders were considered to have good mental health if they scored 18 or less [39].

### 2.3. Data Analysis

A cumulative AYE score ranging from 0 to 7 was calculated for each respondent by adding up the individual score for each AYE item for that respondent. Similarly, a cumulative PYE score ranging from 0 to 10 was calculated. Then, following the methodology employed by Bethell et al. [5] and based on the sample’s distribution, PYE cumulative scores were categorised into groups of 0 to 6 (low), 7 to 8 (moderate) and 9 to 10 (high) PYEs, and AYE cumulative scores were categorised into groups of 0, 1, and ‘more than 1’.

Data selection, variable creation and statistical analysis were completed using SPSS 29.0.2.0 (IBM Corp., Armonk, NY, USA). Initial analyses of AYEs, PYEs, demographic characteristics and outcome variables were completed using statistical tests for ordered and binary data types as appropriate. These included Spearman’s rho (S.rho), chi squared test of independence, Mann–Whitney U (MW-U) and Kruskal–Wallis (KW) tests. Subsequent analysis was performed using logistic regression to elucidate the effects of AYEs, confounders and PYEs on each outcome. Three models were developed: (A) unadjusted univariate logistic regression of AYE score and outcome variable, (B): logistic regression of AYE score and outcome measures adjusted for school sector, sex, indigenous status, rurality, and highest parental income; and (C) multivariate logistic regression of AYE score and PYE score, adjusted for covariates as in model B. No sample or drop out weights were applied during the analysis of these data. This analysis reports solely on the association between adolescent factors and outcomes in adulthood for this sample of individuals who completed all four survey points.

## 3. Results

The data for 3708 young people who participated in all four surveys used in this analysis were extracted, representing 36 percent of the full LSAY03 cohort. Overall, the response rates for the LSAY dropped 2 to 10% per year, with 90.4% of the original cohort completing wave 2, 83.8% completing wave 3 and 36.1% completing wave 10 (25 years). Table 1 shows the key demographics of the full LSAY03 sample at recruitment at age 15, and the sample that participated in all four surveys used in this study. The participants in the traditionally more vulnerable groups (e.g., single-parent family, Indigenous) showed higher drop off rates compared to the less vulnerable individuals. While the study sample is not representative of the full LSAY03 cohort, according to Australian census data, the sample used in this study remained reasonably representative of the 15–19-year-old population at that time [40], noting that parental education is not collected in census data, and that the LSAY data records the school rather than residential (census) location. Nonetheless, the proportion of LSAY sub-sample participants from remote locations was in line with the Australian national census population of 15–19 year olds living remotely (2.1%).

Five AYE domains were identified at age 15–17, consisting of seven factors (Table 2). Overall, the majority of the sample did not experience adversities. The most commonly reported AYE was being discontented with money and/or standard of living (18.3%), followed by living in a single-parent family (16%). The least commonly reported was ‘not living at home or with other relatives’ (5.2%). The number of AYEs reported ranges from 0 to 7, with the median being 1 (IQR 0–2).

Ten positive experiences could be identified that mapped onto three of the four building blocks of HOPE. Overall, this sample had a high prevalence of PYEs, with most PYEs being reported by over 90% of the sample. Two PYEs had noticeably lower rates; these were ‘having two or more non-parent, non-teacher adults important to me’ and ‘participating in an out-of-school activity at least monthly’. The number of PYEs reported ranged from 1 to 10, with a median of 8 (IQR 7–9).

Most of the young people in this sample had positive economic outcomes at age 25, with 88% being engaged in education, training or satisfactory employment (defined as employment that was sufficient to preclude receipt of a welfare payment), 90% reporting good general health and 95% reporting good mental health (Table 2).

Strong ordinal relationships were seen between the number of both PYEs and AYEs and all four outcome measures. A higher number of AYEs was associated with a lower chance of completing post-school education (standardised Mann-Whitney U test statistic (Std. MWU) = −11.1), a lower chance of being in satisfactory employment or training (Std. MWU = −6.6) and a higher chance of having poor general (Std. MWU = −5.3) or mental health (Std. MWU = −5.2) (all *p* < 0.0001). Conversely, having a higher number of PYEs was significantly associated with having a positive outcome; being in satisfactory employment or training (Std. MWU = 3.5); completing post-school education (Std. MWU = 5.2); and good general (Std. MWU = 5.3) or mental health (Std. MWU = 4.5) (all *p* < 0.0001). Figure 1 shows the mean AYE and PYE scores for those having and not having each outcome at age 25.

To address the hypothesis that having a high number of PYEs could mitigate or buffer the negative impact of AYEs, a set of three models was used (Table 3). These assessed (A) the association between the AYE score and outcome, (B) the association between the AYE score and outcome after adjusting for school sector, sex, Indigenous status, rurality, and highest parental income and, (C) the association between the AYE score, PYE score and outcome after adjustment for demographics.

The results of the education and employment outcomes are similar. In both cases, the number of AYEs is strongly negatively associated with the outcome (model A), even after adjustment for key demographics (model B). Model B shows that for each increase in the AYE score, the odds of obtaining further education decrease by 32%, and the odds of being in satisfactory employment, education or training decrease by 30%. The inclusion of PYEs into the model makes little difference to the strength of these associations between the AYEs and outcome (model C).

For both health outcomes, there was a strong negative association between the AYE score and outcome. After adjustment for confounders, it was found that for each increase in the AYE score, the odds for having good health decrease by 23% and the odds of having good mental health decrease by 31%. Unlike in the education and employment outcomes, the inclusion of PYEs into the health outcome models decreased the strength of this association between the AYEs and outcome and significantly improved the model fit. With the inclusion of the PYEs (model C), it was found that for each increase in the AYE score, the odds for having good health decrease by 18% and the odds of having good mental health decrease by 24%. For each increase in the PYE score, the odds for having good health increase by 20% and the odds of having good mental health increase by 28%.

## 4. Discussion

We used a longitudinal dataset to explore the effects of adolescent experiences and environments on young adult health, education and employment. The LSAY03 sample used here was reasonably representative of the population when compared with 2011 census data in terms of gender, Indigenous status and school sector. There was an over-representation of adolescents living in nuclear, two-parent families. The LSAY03 sample at age 25 years was also more likely than young adults in Australia to have post-secondary school qualifications and be employed. This sample could thus be considered to be somewhat more advantaged than the Australian population of young people at that time. Not surprisingly then, only a small proportion of the participants in the LSAY03 cohort reported experiencing living in poverty, having unstable housing, feeling marginalised or having carer responsibilities at the age of 17, and reported overall low numbers of AYEs and high numbers of PYEs.

The LSAY03 data concerning AYEs did not map directly to prior studies of adults’ memories of their own childhood. The survey respondents were minors at the time of the initial survey completion; therefore, questions concerning child abuse and neglect, parental violence and parent/carer mental health were not the focus of the LSAY nor were included in the survey. Nevertheless, having a higher number of the AYEs described here (poor standard of living, single-parent home, experience of homelessness/transience, alienation and carer responsibilities) did have profound negative effects on young adult health and wellbeing, as well as young adult participation in education, training and employment. These findings are consistent with the emerging understanding of the importance and impact of adolescence as a critical developmental period and early adulthood as a key life period of transition [28].

We were able to conceptually map the LSAY03 data to the PYEs related to three of the four building blocks of HOPE, which provided a useful framework for identifying PYEs in the dataset. There were no LSAY03 questions that reported on emotional growth. This may be due to the focus of the survey on education- and employment-related factors but may also represent a failure in the LSAY’s design to acknowledge adolescence as a period of important emotional growth and development. Further, the longitudinal study by Guo, et al. [34] excluded the domain of emotional growth expressed as ‘learning social and emotional competencies’, as they determined these to be a potential outcome of positive experiences. Nevertheless, having a higher number of PYEs within the three building blocks of HOPE, relationships, environment and engagement was significantly positively associated with all the outcomes for young people.

However, when assessed for the impact of PYEs on mitigating AYEs and adjusted for demographic factors (school sector, sex and Indigenous status, rurality and highest parental income), the impacts of the PYEs on the outcomes were only present for general and mental health. This is consistent with the evidence of the impact of material deprivation and the quality of peer relationships in adolescence on health and life satisfaction in adulthood [30,31,32]. There was no evidence of PYEs impacting the relationship between AYEs and the young people completing post-secondary school education. Similarly, the PYEs did not buffer the impact of the AYEs on being engaged in education, training or satisfactory employment at age 25, a result that is not consistent with evidence from other studies that looked at the associations of specific adversities on education and employment outcomes [29,30].

PCEs have been demonstrated to buffer the impact of ACEs on mental health [5]. This impact was seen here on both mental and general health, but not as strongly as in other studies, with both the AYEs and PYEs independently influencing these early adult outcomes. This study did not measure the ACEs usually assessed: child abuse and neglect and household dysfunction [1]. However, these unmeasured ACEs and PCEs could likely be impacting both adolescent experiences and adult outcomes. This study, therefore, is limited to the added impacts of positive and adverse adolescent experiences and demonstrates that the positive and adverse experiences in adolescence both have an impact on early adult outcomes. A limitation of the extant evidence on adverse and positive childhood experiences (ACEs and PCEs) and their relationship to adult outcomes has been that causality cannot be inferred from such cross-sectional surveys, and those studies of ACEs and PCEs may have issues with recall bias. This longitudinal study, in having similar findings based on measuring experiences specifically relating to young people, helps validate the relationships between adverse and positive early life experiences and outcomes. Further, this study based on the Longitudinal Studies of Australian Youth, together with the study by Guo and colleagues using the Longitudinal Study of Australian Children [28], begins to provide evidence on how different types of adverse and positive experiences at different ages can have individual (or potentially cumulative) impacts over time on adult outcomes. Future research should continue to explore this using longitudinal studies to inform policy and actions to minimise the adverse and maximise the positive experiences that matter most at different ages.

In generalising these findings to current contexts, this study was limited by the use of data from 2003 to 2013. This was necessary as items conceptually relating to PYEs and AYEs were removed from the surveys for subsequent LSAY cohorts. Further, the sample was presented as somewhat more advantaged than the general population and may have been impacted by the data collection being limited to those who were in school at the time the cohort commenced and for whom data were available at all four time points. Although this is a relatively large longitudinal dataset, the statistical analysis was limited for less common exposures or outcomes. Whilst the findings on general and mental health are likely to be consistent over time (COVID-19 pandemic impacts not withstanding), educational and employment outcomes are likely subject to policy and economic circumstances that change over time, suggesting caution is warranted in applying these results to current circumstances.

Finally, these data were not intended for measuring AYEs and PYEs, and so were conceptually mapped to the literature-informed concepts and building blocks of HOPE, particularly noting that no items related to emotional growth were available. Some items, such as carer responsibilities, did not provide enough differentiation to be clearly classified as adverse or positive experiences; however, where there was a lack of nuance, the balance of previous evidence on the negative impacts on young adults of having carer responsibilities frequently each week informed its classification as adverse [37].

This study has demonstrated that adverse and positive experiences reported by adolescents about their lives in real time impact on their adult outcomes without the limitation of recall bias. This study adds to the growing body of research emphasising the importance of adolescent neurodevelopment and young adulthood as a key time of transition. Importantly, this study demonstrated that positive experiences in adolescence significantly impact adult outcomes. Generally, research in this area has a deficit perspective [30,31,32], viewing adolescence as a window of opportunity to ‘fix’ what was broken during childhood. This is one of the first studies to use a strengths-based, health promotional perspective to identify how to improve adult outcomes for all adolescents though promoting positive relationships; safe, equitable and stable environments; and social and civic engagement.

## 5. Conclusions

This study demonstrates the strength of using prospective longitudinal study to complement adult recall data to explore the impact of adverse and positive experiences in the lives of young people. This study looked at a broad range of outcomes, including employment and education, as well as general and mental health, which reflect successful transition to adult life. The findings further demonstrated that the evidence of the relationships between adverse experiences, positive experiences and outcomes is robust, and holds true for adolescents as they transition to adulthood. This transition, therefore, is another point in the life course where attention should be paid to maximising positive and minimising adverse experiences as these also have an impact on adult outcomes. Future prospective longitudinal cohort studies should look at adverse and positive childhood experiences, as well as Adverse and Positive Youth Experiences, to fully understand the multiple windows of opportunity to maximise adult health and wellbeing.

## Figures and Tables

**Figure 1 ijerph-21-01147-f001:**
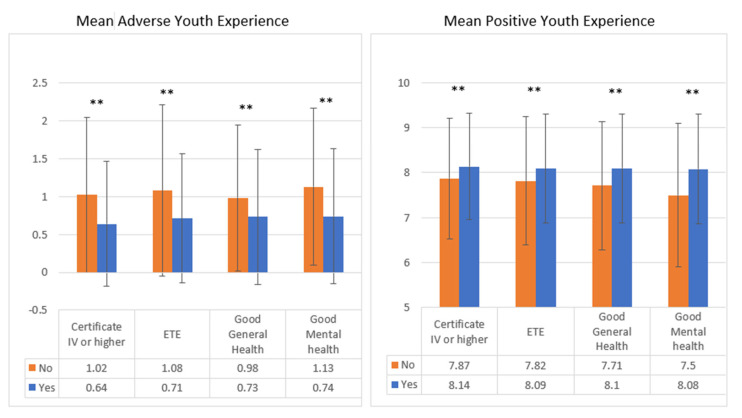
Mean Adverse and Positive Youth Experience score by outcome measure. ** Mann-Whitney U analysis, *p* < 0.0001.

**Table 1 ijerph-21-01147-t001:** Sample demographics.

Demographics		LSAY				Census 2006
		Wave 1 (*n* = 10,370)		Waves 1, 2, 3 and 10 (*n* = 3708)		
		*n*	%	*n*	%	
Sex of respondent	Male	5149	49.7	1848	49.8	51.3
	Female	5221	50.3	1860	50.2	48.7
Indigenous status	Non-Indigenous	9781	94.3	3594	96.9	96.4
	Indigenous	589	5.7	114	3.1	3.6
Family structure	Single-parent family	2071	20.0	584	15.7	25.1
	Nuclear family	7149	68.9	2821	76.1	72.1
	Mixed family	830	8.0	221	6.0	-
	Other	256	2.5	64	1.7	2.7
	Missing	64	0.6	18	0.5	-
School sector	Government	6643	64.1	2186	59.0	61.8
	Catholic	2135	20.6	803	21.7	21.5
	Independent	1592	15.4	719	19.4	16.7
Highest level of parental education *	Low	1304	12.6	342	9.2	
	Medium	3300	31.8	1011	27.3	
	High	5527	53.3	2299	62.0	
	Missing	239	2.3	56	1.5	
Location	Metropolitan	7300	70.4	2697	72.7	
	Urban	1539	14.8	476	12.8	
	Suburban	1305	12.6	473	12.8	
	Remote	226	2.2	62	1.7	

* International Standard Classification of Education (ISCED), rating < 3 = low education, rating of 3 or 4 = medium education, rating > 4 = high education.

**Table 2 ijerph-21-01147-t002:** Prevalence of positive and adverse experiences between ages 15 and 17, and outcome measures at age 25.

Domain		Factors	No		Yes	
Adverse Youth Experiences			*n*	%	*n*	%
Living in poverty	1	Low number of possessions	3274	88.3	434	11.7
	2	Discontented with money/standard of living	3023	81.5	685	18.3
Single parent	3	Single-parent family	3124	84.3	584	15.8
Unstable housing	4	Not living at home or with other relatives	3650	98.4	58	5.2
Marginalised	5	Not belonging at school	3486	94.0	222	6.0
	6	Not treated fairly by teachers	3365	90.7	343	9.3
Carer responsibilities	7	Frequently look after other people	3242	87.4	466	12.6
Positive Youth Experiences			*n*	%	*n*	%
Good relationships	1	Close friends are very important	132	3.6	3576	96.4
	2	Household members are important	175	4.7	3533	95.3
	3	Two or more groups of (non-teacher/non-household) adults are important to me	2005	54.1	1713	45.9
	4	Teachers listen and/or help me	249	6.7	3459	93.3
Environment	5	Belong at school	268	7.2	3440	92.8
	6	Good student–teacher relationships at school	285	7.7	3423	92.3
	7	Feel safe at school	172	4.6	3536	95.4
	8	Feel safe in neighbourhood	582	15.7	3126	84.3
Social engagement	9	School activity at least monthly	880	23.7	2828	76.3
	10	Out-of-school activity at least monthly	2455	66.2	1253	33.8
Outcome measures			*n*	%	*n*	%
Employment	Employed FT or PT		401	10.8	3307	89.2
	Not receiving a government payment		469	12.6	3239	87.4
In education or training	Currently in FT or PT education		2818	76.0	890	24.0
In education, training or satisfactory employment	1	(Employed FT or PT AND not receiving a government payment) AND/OR (Currently in FT or PT education)	439	11.8	3269	88.2
Post-school education	2	Cert IV or above (post-school trade orequivalent qualification)	1077	29.0	2631	71.0
General health	3	Good to excellent general health	387	10.4	3321	89.6
Mental health	4	Kessler score indicated good to excellentmental health	167	4.5	3541	95.5

FT = full-time employment, PT = part-time employment, Cert IV = certificate IV.

**Table 3 ijerph-21-01147-t003:** Univariate and multivariate logistic regression models showing the interrelationship between adverse and positive experiences in adolescence and long-term education, employment and health outcomes.

Model		Certificate IV or Higher				Education, Training, or Satisfactory Employment			
	Variable	Wald	Sig.	Exp (B)	95% C.I.	Wald	Sig.	Exp (B)	95% C.I.
A	AYE score	125.51	<0.001	0.64	(0.59 to 0.69)	61.85	<0.001	0.67	(0.61 to 0.74)
B	AYE score	84.24	<0.001	0.68	(0.63 to 0.74)	44.78	<0.001	0.71	(0.64 to 0.78)
C	AYE score	74.09	<0.001	0.69	(0.63 to 0.75)	36.86	<0.001	0.72	(0.65 to 0.80)
	PYE score	1.33	0.25	1.04	(0.98 to 1.10)	2.07	0.15	1.06	(0.98 to 1.15)
Model		Good General Health				Good Mental Health			
	Variable	Wald	Sig.	Exp (B)	95% C.I.	Wald	Sig.	Exp (B)	95% C.I.
A	AYE score	26.27	<0.001	0.76	(0.68 to 0.84)	29.10	<0.001	0.67	(0.58 to 0.78)
B	AYE score	22.61	<0.001	0.77	(0.69 to 0.86)	23.53	<0.001	0.69	(0.60 to 0.80)
C	AYE score	11.94	<0.001	0.82	(0.73 to 0.92)	12.10	<0.001	0.76	(0.65 to 0.89)
	PYE score	18.17	<0.001	1.20	(1.10 to 1.30)	16.87	<0.001	1.28	(1.14 to 1.44)

Models: (A) Unadjusted univariate logistic regression of AYE score and outcome variable. (B) Logistic regression of ACE score and outcome measure adjusted for school sector, sex, Indigenous status, rurality, and highest parental income. (C) Multivariate logistic regression of AYE and PYE scores adjusted for all covariates in model B. Sig: significance level with alpha = 0.05. Exp (B): Exponential of Coefficient (B), also known as the Odds Ratio. 95% C.I.: 95% confidence interval for the Exp (B).

## Data Availability

The original data presented in the study are openly available in Australian Data Archive (ADA) at the Australian National University at https://doi.org/10.4225/87/5IOBPG. Access to the data is free via a formal request and registration process managed by the ADA.

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
