# Peer review of "The Impact of Positive and Adverse Experiences in Adolescence on Health and Wellbeing Outcomes in Early Adulthood"

_ijerph, 2024, doi:10.3390/ijerph21091147_

Round 1

Reviewer 1 Report

Comments and Suggestions for Authors

While the manuscript presents interesting findings on the impacts of adverse youth experiences and positive youth experiences, it suffers from several methodological and interpretative weaknesses that cannot be remedied through revisions. Furthermore, considering that the study does not even offer highly pioneering insights, its contribution to the literature appears limited.

Main weaknesses of the manuscript include the following: the sample is not representative of the LSAY cohort; the measures employed are not validated; the theoretical framework is insufficiently robust, notably omitting a crucial domain (emotional growth); the data utilized is relatively outdated, spanning 11 to 21 years; the classification of carer responsibilities as adverse experiences lacks necessary nuance; and key variables such as child abuse, neglect, and parental violence were not included in the surveys, limiting the comprehensiveness of the AYE data.

Author Response

Responses to reviewer comments

The impact of positive and adverse experiences in adolescence on health and wellbeing outcomes in early adulthood

Reviewer comments

Author’s responses

Reviewer one

While the manuscript presents interesting findings on the impacts of adverse youth experiences and positive youth experiences, it suffers from several methodological and interpretative weaknesses that cannot be remedied through revisions.

Please see responses to reviewer individual concerns below.

Furthermore, considering that the study does not even offer highly pioneering insights, its contribution to the literature appears limited.

Text has been added to the introduction and discussion highlight the contribution this paper makes to the literature.

Main weaknesses of the manuscript include the following:

·         the sample is not representative of the LSAY cohort;

We acknowledge in the article that the sample is not representative of the original LSAY cohort, due to attrition. However, we have added Australian Census data to Table 1 demonstrating the sample used is representative of the population at that time. We have also noted this in the manuscript.

·         the measures employed are not validated;

Please note that this was a secondary analysis of pre-existing data. We have added text to the methods to make this clearer.

·         the theoretical framework is insufficiently robust, notably omitting a crucial domain (emotional growth);

Please note that this was a secondary analysis of pre-existing data that did not include these items. This has been made clearer in the method and is acknowledged in the discussion.

·         the data utilized is relatively outdated, spanning 11 to 21 years;

Please note that this was a secondary analysis of pre-existing data. The age of the data is acknowledged in the discussion, and the rationale for its use is provided in the methods. 

·         the classification of carer responsibilities as adverse experiences lacks necessary nuance;

Thank you, this is acknowledged in the discussion as a limitation.

·         and key variables such as child abuse, neglect, and parental violence were not included in the surveys, limiting the comprehensiveness of the AYE data.

We have added text to the introduction, methods and discussion to make the rationale and value of this study, despite the absence of these data usually considered as ACEs clear. We believe that in considering the further adversities that can be experienced in adolescence, this paper makes an important contribution.

Reviewer 2 Report

Comments and Suggestions for Authors

This paper is very well written, the langage is clear, coherent and easy to read.

All the arguments are well referenced in the references.

This paper aims at looking at the buffering effect of positive youth experiences (PYE) on Adverse Youth experiences (AYE).

The results are supporting the hypotheses : the odds of having good mental health at age 25 decrease by 31% for each AYE score increase.

However, when taking into account PYE in the statistical model, the odds of having good mental health at age 25 is deacreased by 24% for each AYE.

This means that PYE buffer reduce the effect of AYE on mental health (and general health too).

However, PYE and AYE were evaluated at the same time, which means that the buffer is effective only when positives youth experience are lived in the same period of time than the negative ones.

This raised a question : does this buffer is still effective when AYE are lived before PYE ? and more importantly, does PYE is still effective if they happen before the adverse ones ? are there data on this topic ?

At last, regarding the scoring of the Kessler Psychological distress measure - LSAY, the cutoff seems to be arbitrary : are there data, or is there a specific reason to choose 18 as a cutof ? this choice is not justified in the manuscript by the authors.

Author Response

Responses to reviewer comments

The impact of positive and adverse experiences in adolescence on health and wellbeing outcomes in early adulthood

Reviewer comments

Author’s responses

Reviewer two

This paper is very well written, the langage is clear, coherent and easy to read. All the arguments are well referenced in the references.

This paper aims at looking at the buffering effect of positive youth experiences (PYE) on Adverse Youth experiences (AYE).

Thank you.

The results are supporting the hypotheses : the odds of having good mental health at age 25 decrease by 31% for each AYE score increase.

However, when taking into account PYE in the statistical model, the odds of having good mental health at age 25 is deacreased by 24% for each AYE.

This means that PYE buffer reduce the effect of AYE on mental health (and general health too).

However, PYE and AYE were evaluated at the same time, which means that the buffer is effective only when positives youth experience are lived in the same period of time than the negative ones.

This raised a question : does this buffer is still effective when AYE are lived before PYE ? and more importantly, does PYE is still effective if they happen before the adverse ones ? are there data on this topic ?

The reviewer has noted a very interesting question here that is not able to be answered by the data available in this study. However, we have noted that this study, together with the study by Guo and colleagues that used longitudinal data in early childhood, can start to provide data that can provide some beginning understanding of what adversities and positive life experiences matter at particular life stages. We have added text to the discussion to reflect this, and noted this as an area for further research.

At last, regarding the scoring of the Kessler Psychological distress measure - LSAY, the cutoff seems to be arbitrary : are there data, or is there a specific reason to choose 18 as a cutof ? this choice is not justified in the manuscript by the authors.

We have added text and references to the method to justify the cohort, in accordance with the Australian scoring rubric for the K6. We note here that the Australian scoring rubric differs from that used in the USA and this may have lead this reviewer to make this comment.

Reviewer 3 Report

Comments and Suggestions for Authors

The manuscript entitled “The Impact of Positive and Adverse Experiences in Adolescence on Health and Wellbeing Outcomes in Early Adulthood” is an interesting and valuable contribution to the literature on the antecedents of well-being and its opposite psychological state. In my modest opinion, the manuscript deserves publication after a few concerns are adequately addressed by the authors. The following are such concerns:

In the introductory section, the authors may want to address more broadly the contribution that the current study intends to make to the extant literature. Namely, what does this study possess that earlier studies have missed?

The theoretical framework is nebulous. How does the study fit with existing models of human development? How do its likely findings fit such models? Consideration of cognitive, emotional, and behavioral dimensions may be helpful.

In the method section, how were the participants’ responses coded for AYE and PYE? What were the interrater reliability values?

In the discussion section, the authors may want to broaden the practical and theoretical implications as well as applications of their findings. 

Comments on the Quality of English Language

As noted earlier, minor proofreading may be desirable.

Author Response

Responses to reviewer comments

The impact of positive and adverse experiences in adolescence on health and wellbeing outcomes in early adulthood

Reviewer comments

Author’s responses

Reviewer three

The manuscript entitled “The Impact of Positive and Adverse Experiences in Adolescence on Health and Wellbeing Outcomes in Early Adulthood” is an interesting and valuable contribution to the literature on the antecedents of well-being and its opposite psychological state. In my modest opinion, the manuscript deserves publication after a few concerns are adequately addressed by the authors. The following are such concerns:

Thank you. Please see responses to your concerns below.

·         In the introductory section, the authors may want to address more broadly the contribution that the current study intends to make to the extant literature. Namely, what does this study possess that earlier studies have missed

Further text has been added to the introduction and the discussion to highlight the contribution of this study and further contextualise the contribution.

·         The theoretical framework is nebulous. How does the study fit with existing models of human development? How do its likely findings fit such models? Consideration of cognitive, emotional, and behavioral dimensions may be helpful.

The findings of this paper resonate with extant literature related to childhood adversity and their deleterious and persistent impact on health and wellbeing. We extend this work by using the HOPE framework as a basis to define and explore the impacts of positive experiences on adult outcomes, which is still a relatively new area of inquiry. The HOPE framework does not lend itself to cognitive and behavioural aspects, and we note that the data set used did not contain data related to fourth building block (emotional growth). We have added a comment in the introduction acknowledging the lack of definitional consensus around what constitutes positive experiences. How this study fits with existing models of human development is beyond the intended scope of this paper, but may serve as a topic of future work.

·         In the method section, how were the participants’ responses coded for AYE and PYE? What were the interrater reliability values?

This information is in the supplemental file – we can include the supplemental table within the paper if the editors consider that would be better. We are not sure what the reviewer is asking with regards to interrater reliability. In terms of deciding how the variables were constructed against the AYE/PYE, some were obvious. For others the research team agreed the coding based conceptually on the HOPE framework. We have added a sentence to the method to make the process of conceptually identifying the variables clearer.

·         In the discussion section, the authors may want to broaden the practical and theoretical implications as well as applications of their findings. 

We have added further comment on the practical implications of the work to the introduction and study, as well as the contribution of the work to further validating the HOPE framework.

Reviewer 4 Report

Comments and Suggestions for Authors

The research is well written. Importance is given to the specific detailing of every section. There is a logical flow seen in the materials and methods, results, and discussion sections of the paper. 

Author Response

Responses to reviewer comments

The impact of positive and adverse experiences in adolescence on health and wellbeing outcomes in early adulthood

Reviewer comments

Author’s responses

Reviewer four

The research is well written. Importance is given to the specific detailing of every section. There is a logical flow seen in the materials and methods, results, and discussion sections of the paper. 

Thank you.

Reviewer 5 Report

Comments and Suggestions for Authors

Dear Authors,

I congratulate you on a successfully written paper.  The work is stylistically well written. However, the content is not adequate, at least in my opinion. From the description of the methodology, I could not understand how you designed the study? What did you do with the data that was already taken in 2003? In that longitudinal study whose data you cited, had they already been processed at that time? What have you done now? It is not clear to me from the text in the scientific paper.

And secondly, I don't understand what the scientific contribution of this study of yours is? Isn't it expected and common knowledge that bad experiences growing up are strongly associated with bad outcomes in adulthood?

What's new in your study?

There are some typos, but the language style is adequate overall.

Taking into account everything I have stated, I believe that the scientific work can possibly be reconsidered after excessive corrections of the methodology and conclusions.

With respect,

Comments on the Quality of English Language

Minor English editing is needed.

Author Response

Responses to reviewer comments

The impact of positive and adverse experiences in adolescence on health and wellbeing outcomes in early adulthood

Reviewer comments

Author’s responses

Reviewer five

I congratulate you on a successfully written paper.  The work is stylistically well written.

Thank you

However, the content is not adequate, at least in my opinion. From the description of the methodology, I could not understand how you designed the study? What did you do with the data that was already taken in 2003? In that longitudinal study whose data you cited, had they already been processed at that time? What have you done now? It is not clear to me from the text in the scientific paper.

We have added text and restructured the methods section to clarify the use of the LSAY, secondary analysis and the data used in this study.

And secondly, I don't understand what the scientific contribution of this study of yours is? Isn't it expected and common knowledge that bad experiences growing up are strongly associated with bad outcomes in adulthood?

What's new in your study?

Further text has been added to the introduction and the discussion to highlight the contribution of this study and further contextualise the contribution.

There are some typos, but the language style is adequate overall.

Taking into account everything I have stated, I believe that the scientific work can possibly be reconsidered after excessive corrections of the methodology and conclusions.

The manuscript has been extensively reviewed and typos corrected. Thank you.

Round 2

Reviewer 1 Report

Comments and Suggestions for Authors

The authors have significantly improved the manuscript since the previous version, and it could now be considered worthy of publication.

Reviewer 5 Report

Comments and Suggestions for Authors

Dear Authors,

Ithank you on applaying my comments. The artice is now more clear and suitable for publication.